# Dietary Habits, Anthropometric Features and Daily Performance in Two Independent Long-Lived Populations from *Nicoya peninsula* (Costa Rica) and *Ogliastra* (Sardinia)

**DOI:** 10.3390/nu12061621

**Published:** 2020-06-01

**Authors:** Alessandra Nieddu, Laura Vindas, Alessandra Errigo, Jorge Vindas, Giovanni Mario Pes, Maria Pina Dore

**Affiliations:** 1Dipartimento di Scienze Mediche, Chirurgiche e Sperimentali, University of Sassari, Viale San Pietro 8, I-07100 Sassari, Italy; alessandramrnieddu@gmail.com (A.N.); gmpes@uniss.it (G.M.P.); 2Asociación Península de Nicoya–Zona Azul, 145-1100 San José, Costa Rica; laura.vindasmeza@ucr.ac.cr (L.V.); jvin63@yahoo.com (J.V.); 3Dipartimento di Scienze Biomediche, University of Sassari, Viale San Pietro 43/b, I-07100 Sassari, Italy; alessandra.errig@tiscali.it; 4Sardinia Longevity Blue Zone Observatory, 08040 Ogliastra, Italy; 5Baylor College of Medicine, One Baylor Plaza, Houston, TX 77030, USA

**Keywords:** longevity, diet, Costa Rica, Sardinia

## Abstract

(1) Background: Longevity Blue Zones (LBZs) are populations characterized by exceptional longevity. The purpose of this cross-sectional study was to compare the food habits of two representative samples of the oldest old subjects from the population residing in the LBZs of Nicoya peninsula (Costa Rica) and in the mountainous part of Ogliastra (Sardinia, Italy). (2) Methods: Data were collected using validated tools, including a food frequency questionnaire, Basic Activities of Daily Living (BADL) and Instrumental Activities of Daily Living (IADL) scales for functional autonomy, body mass index, and waist and limbs circumferences. (3) Results: A total of 210 subjects, 60 (31 male) from Nicoya (age range 80–109 years), and 150 (61 male) from Ogliastra (age 90–101 years) were included in the study. In both populations, the highest frequencies of consumption were recorded for plant-derived foods (cereals 60–80% daily, legumes ≥ 80% daily in Nicoya, ≥ 60% 2–5 servings/week in Ogliastra), followed by those of animal origin (dairy products, meat) ≥60% and 80% daily, in Nicoya and Ogliastra, respectively. The frequency of milk consumption showed a positive correlation with BADL (ρ = 0.268 for Nicoya and ρ = 0.214 for Ogliastra) and IADL scores (ρ = 0.466 for Nicoya and ρ = 0.471 for Ogliastra), whereas legumes consumption correlated negatively with self-rated health (ρ = −0.264) and IADL (ρ = −0.332). (4) Conclusions: Our results indicate that the dominant dietary model among the elderly of Nicoya and Ogliastra is a plant-based diet complemented by a non-negligible consumption of animal products, mostly dairy products. Further prospective studies are needed to ascertain a possible cause–effect relationship between food habits and increased likelihood of reaching advanced age.

## 1. Introduction

Longevity Blue Zones (LBZs) are geographical areas of the planet harboring a measurably higher proportion of long-lived individuals compared with the usual average elsewhere [1,2]. They are mostly isolated areas, such as the island of Okinawa in Japan, the peninsula of Nicoya in Costa Rica, the island of Ikaria in Greece, and the Ogliastra subregion on the Mediterranean island of Sardinia, all well known for hosting a large number of age-validated centenarians [3]. In these populations exceptional longevity appears as a widespread phenomenon, which explains the term “population longevity” proposed in contrast to that of “individual longevity” [4]. The analysis of LBZs has been claimed to facilitate the identification of longevity factors acting at the superindividual level (for a review, see [2,3,4]). A number of hypotheses have been posed to explain the emergence of these long-lived populations, specifically targeted on genetic factors, environment, lifestyle, occupational activity, and social life [5]. Among the lifestyle-related factors, dietary habits play a fundamental role: It is known that the ability of some individuals to reach an advanced age, by escaping most chronic non-transmissible diseases, derives at least in part from adopting a healthy diet for most of their lives.

Various food styles share the merit of promoting human longevity. Among them, the Mediterranean diet is usually considered one of the most capable of protecting against age-related disease, ensuring longer survival [6], although the exact mechanism(s) by which the adherence to this dietary model acts positively are poorly known [7] or controversial [8]. Furthermore, the study of the relationship between the consumption of specific foods and health status does not yet allow definitive conclusions. Most of the putative beneficial aspects of the Mediterranean diet, widely publicized in popular literature, are generally attributed to the content of antioxidants [9], the low consumption of processed meat [10], and the consumption of olive oil [11] and/or red wine [12].

Ideally, a comparison between the dietary pattern of the elderly population of all four LBZs recognized so far would contribute to shed light on the role of diet in promoting successful aging. Yet this comparison is not easily achievable for various reasons, for example, the Ikaria population is numerically too small to be informative. On the other hand, the Okinawa population is quite different in terms of genetic structure, food history, ethnic characteristics and lifestyle from that of the other three LBZs, making the interpretation of such comparative analysis too complex. A realistic comparison is therefore feasible only between the population of Nicoya, Costa Rica, and Ogliastra in Sardinia, Italy. Besides, the cultural heritage of the two populations was historically influenced by Spanish rule, which may have left common traces in the food traditions [13].

The main objective of this study was to evaluate comparatively the eating habits and functional ability of elderly people living in two LBZs, the Nicoya peninsula and Ogliastra.

## 2. Materials and Methods 

### 2.1. Study Populations

#### 2.1.1. Nicoya, *Costa Rica*

The Nicoya peninsula is a north-western region of Costa Rica (Central America) facing the Pacific Ocean at latitude North of 10° 04′ and longitude West 85° 25′. It encompasses the province of Guanacaste and hosts a population of about 326,000 inhabitants spread over an area of 778 km² [14]. The area was once covered with tropical dry forests until it was converted into pastureland during the 1950s [15]. The peninsula of Nicoya, compared to Costa Rica as a whole, appears to be a relatively isolated region (the Tempisque river separates it from the rest of the country), a characteristic shared by the other three LBZs [2]. The ethnic composition of the population in the peninsula is slightly different from the whole of Costa Rica. Whereas in the latter the indigenous population is less than 2%, in the Nicoya peninsula this rises to 5% and includes, in particular, the Chorotega ethnic group, descendants of Native Americans whose traditions may have had some influence on the lifestyle and current behavior of the Nicoya population [16].

In 2008, the demographer Luis Rosero-Bixby reported a life expectancy at birth of 76.2 years among the male population of Nicoya, i.e., higher than that of 74.8 recorded for white males in the United States [17]. In this region, the overall mortality rate was 20% lower than that of the remaining part of the country; the proportion of people over 65 years of age was 8% and, as in other LBZs, economic indicators show a modest degree of underdevelopment compared to the rest of the country. The advantage of Nicoya’s males in terms of survival is absent in females, but also tends to disappear in emigrants abroad [17], and according to the CRELES study is essentially due to a lower incidence of cardiovascular disease [18]. This investigation, which also considered some bio-medical markers, showed that the average body height of the inhabitants of Nicoya was higher than the general population of Costa Rica, and their body mass index was lower, as was the prevalence of physical and mental disability [18]. The leukocyte telomere length, considered an indicator of longevity, is higher among Nicoya’s inhabitants [19]; however, this trait also depends on factors related to lifestyle, such as stress and daily physical activity, and does not exclusively reflect the individual genetic makeup [20].

The functional and social profile of centenarians from the Nicoya peninsula has recently been described by Madrigal-Leer et al. [21] and the eating habits of 18 men and 16 women between the ages of 90 and 109 have been analyzed by Momi-Chacón et al. [22]. The latter study revealed a predominantly plant-based diet characterized by low consumption of red meat and a lower daily calorie intake than in the rest of Costa Rica.

#### 2.1.2. Ogliastra, Sardinia

The central-eastern part of Sardinia, Italy, corresponding to the historical subregions of Ogliastra and Barbagia, at latitude North of 39° 55′ and longitude East 9° 31′, was among the first high-longevity areas to be identified worldwide [23] during a comprehensive validation process of the age of Sardinian centenarians [24,25]. An editorial appeared in 2001 in the Science journal that acknowledged the genuineness of the Sardinian phenomenon and outlined that the male population was especially involved [26]. The longevity area encompasses a group of six municipalities (Arzana, Baunei, Seulo, Talana, Villagrande Strisaili and Urzulei) located around the central mountain, Gennargentu [5,24]. The total population (currently 12,000 inhabitants) is mainly engaged in agricultural activities, maintaining a relatively traditional lifestyle. The causes of longevity in the Sardinian LBZ are currently the subject of intense scrutiny involving several disciplines. Early studies hypothesized a strong impact of genetic factors; however, the results of the AKEA study, which analyzed the transmission of parental longevity in 204 Sardinian centenarians [27] did not reveal a significant vertical transmission of longevity in both the paternal and maternal lineage. In addition, genetic association studies have been performed on the oldest Sardinians, using markers notoriously associated with longevity; however, in terms of frequency, none of these markers have been shown to be significantly different from that of the general Sardinian population [28]. Behavioral and socio-cultural factors, such as nutrition, physical activity and family support, seem to be relatively more important [2,5].

### 2.2. Study Design

This cross-sectional study was conducted with 210 subjects (118 females and 92 males) aged between 80 and 109 years belonging to the Nicoya and Ogliastra populations, respectively. More specifically, the analyses included, (i) an evaluation of dietary habits; (ii) determination of some anthropometric indexes (body weight and height, body mass index (BMI), waist and limb circumferences) taken as long-term markers of nutritional status; and (iii) an evaluation of performance in daily activities.

### 2.3. Study Participants

The first sample consisted of 60 individuals who lived in the Nicoya peninsula and were examined directly by two of the authors (A.N. and L.V.) between July and August 2019. The identification of the subjects, extracted from an electronic database, was provided by the Asociación Península de Nicoya-Zona Azul. Inclusion criteria were Nicoyan origin (to ensure genetic homogeneity) and age of at least 80 years. Every Costa Rican citizen has, since his/her birth or the date of possible naturalization, an identification number reported on the *cédula* (identity card). To minimize error, the age of participants was double-checked through the Costa Rican electoral lists (*padrón*) that are updated every four years and made available by the Tribunal Supremo de Elecciones.

The second sample consisted of 150 subjects from Ogliastra, recruited during an ongoing study, the design and preliminary results of which were previously reported [5]. Inclusion criteria were similar to those used for Nicoya. Age validation was performed as previously described [25].

### 2.4. Data Collection

The interviews at the participants’ home were conducted in Spanish and Italian language to make participants feel comfortable. Information about food and daily activities were recorded, and in the case of subjects with moderate/severe dementia, family members were involved in the interview. Sociodemographic information recorded included age, marital status, education, smoking habits, and living conditions.

Body height was measured in centimeters using a stadiometer, with the patient’s head aligned according to the Frankfurt horizontal plane, and body weight was measured using an electronic scale with an accuracy up to 0.1 kg [29].

The number of variables related to functional status was restricted to the functional ability in daily life, which has better relevance for quality of life measurement. Self-rated health was evaluated according to Idler at al. [30]. Performance-based functional ability was tested using two validated tools: (i) The Basic Activities of Daily Living (BADL), which evaluates the autonomy in carrying out the activities of everyday life, through 6 domains (washing, dressing, using the bathroom, eating, checking urinary and intestinal functions, performing small movements) [31]; and (ii) a modified version of Instrumental Activities of Daily Living (IADL), which estimates the autonomy in performing instrumental activities that are physically and cognitively more complex necessary for an independent life [32]. Only three items were used in this study: telephone use, TV watching and use of money [33]. 

### 2.5. Nutritional Evaluation

Information about dietary habits was collected using a simplified food frequency questionnaire (FFQ) already used for nutritional studies in Sardinia [34] and adapted for the Costa Rican population, which included the following common foods: Meat, fish, vegetable, legumes, cereals, potatoes, pasta, sweets, milk, other dairy foods including soft and hard cheese, and coffee. A list of foods typical of the Nicoya and Ogliastra areas was also included. Nutritional evaluation was complemented by anthropometric measurements, using standardized methods [35], and included the waist circumference (measured at the umbilical scar); the average circumference of the calf (in correspondence to the gastrocnemius muscle belly); the brachial circumference (in correspondence with the biceps muscle belly), as well as the knee-floor distance (measured from the patella lower edge to the floor) [35]. Body mass index was calculated by dividing body weight (in kilograms) by height (in meters) squared (kg/m²).

### 2.6. Statistical Analysis

The basic descriptive statistics for each variable in both populations were reported as the mean and standard deviation for the continuous variables and the absolute and percentage frequencies for the categorical variables. Marital status was labelled as single, married or free union, widowed, and divorced. Education was expressed as the number of years spent at school, which in some developing populations is more informative than the achieved school level. Living conditions were expressed as two categories: (i) Residents in owned or leased houses, or (ii) residents in nursing homes. Smoking was stratified as (i) Non-smokers, (ii) former smokers, and (iii) current smokers. Intake of food categories was coded into an ordinal variable: (i) Never/rarely, (ii) 2–3 servings/month, (iii) 1–2 servings/week, (iv) 3–5 servings/week, and (v) every day. The BMI was divided into <18 (kg/m²), 18–24.9 (kg/m²), 25−29.9 (kg/m²), and ≥30 (kg/m²). The score obtained in the BADL and IADL was stratified into “severe disability” (score < 3) and “moderate disability or no disability” (score ≥ 3). The differences between the two populations were further analyzed by two-tailed Mann–Whitney *U* test for independent samples and the chi-square test for categorical variables. Correlation analysis was performed by calculating the Spearman correlation coefficient. All statistical analyses were performed using SPSS software (version 16.0, Chicago, IL, USA). *p*-values < 0.05 were considered statistically significant. The study was conducted in accordance with the Helsinki Declaration.

## 3. Results

### 3.1. Descriptive Statistics

#### 3.1.1. Sociodemographic Variables

Table 1 shows the differences between the two ethnic groups examined. The mean age of the participants at the time of recruitment was significantly higher in the Costa Rican participants compared to the Sardinian participants. The oldest subjects interviewed were a 109 year-old female and a 106 year-old male from Nicoya. The proportion of unmarried subjects was higher among Sardinians, mainly among females, while married status was more frequent in Costa Rican participants, especially among males. The level of education was very low in both ethnic groups, principally in females from Nicoya. The majority of the elderly at the time of interview lived in their own or a rented dwelling, and less than 6% lived in nursing homes, with no substantial differences between Nicoyans and Sardinians. A smoking habit was detected in 87.4% of the Costa Rican males while none of the Costa Rican females were current smokers, although 17.2% were former smokers, a proportion seven times higher than in Sardinian females.

#### 3.1.2. Anthropometric Parameters

As reported in Table 2 the average weight did not show significant differences between the two ethnic groups or sexes. The average body height in Nicoya’s elders showed values significantly higher than those recorded in Sardinians, both in males and females. The distribution of BMI categories revealed that the elderly from Nicoya were less frequently overweight compared to their Sardinian peers, and there was a greater frequency (30%) of underweight Costa Rican females compared with the males from the same area. Anthropometric measures were overall similar in the two ethnic groups with the notable exception of the knee–floor distance, which showed higher average values in the Costa Ricans than in the Sardinians (nearly 8 cm difference) in both sexes. No significant differences were detected in limb circumference.

#### 3.1.3. Self-Rated Health and Performance-Based Functional Capacity

The results of the health status evaluation of the Costa Rican and Sardinian elders are listed in Table 3. The average self-rated health score was slightly greater among Sardinian males, although the difference was not statistically significant. The level of functional capacity detected severe disability (BADL score <3) in a quarter of the respondents from both ethnic groups, to a greater extent in females than in males, as expected.

#### 3.1.4. Frequency of Consumption of Some Foods

The average frequency of food servings is shown separately in males and females in Figure 1. The foods most often consumed in Nicoya, with equal frequency in both sexes were legumes, cereals, fruit, milk, coffee, and sweets. Dairy foods were less frequently consumed by females, and fish was less frequently consumed by males, although without a statistically significant difference.

A comparison between the food consumption frequencies of elders from Nicoya and those from Ogliastra shows that cereals, salad, meat, milk and sweets consumption display a very similar trend (Figure 1). Contrary to expectations, potato consumption was very high in both populations with a slightly higher intake among Sardinians. Significant differences were observed for sea food, which was consumed very rarely compared to Nicoya’s inhabitants. Among Sardinians the consumption of pasta and dairy products (especially aged cheese) was greater than among Nicoyans, as expected; however, they notably consumed legumes and fruit less frequently than Nicoyans. As for the differences between males and females, no difference was highlighted in both populations, with the exception of coffee among Sardinians females, who on average consumed it less than males.

Appendix A shows the frequencies of consumption of some foods characteristic of the Nicoyan area [13]. It can be noticed that in both females and males, four typical foods were consumed with a particularly high frequency, namely, in females *Gallo Pinto* (every day in 79.3%), tortillas de maíz (69.0%), *Cuajada* (62.1%), and *Gallo Pinto con huevos* (55.2%); and in males *Gallo Pinto* (86.7%), tortillas (76.7%), *Cuajada* (50.0%), and *Gallo Pinto con huevos* (46.7%). It is interesting to note that among the alcoholic beverages, *Guaro* was practically never consumed by females, while in males it was not consumed more than 2–3 times per month. Other fermented beverages such as *Chicha* and *Chicheme* were also occasionally consumed. Appendix A shows the frequency of some typical foods of Ogliastra. Although no comparison can be made with Appendix A, it can still be noted that some foods considered particularly traditional are still consumed by the elders with a non-negligible frequency. In particular, *Pistokku bread* is consumed daily (100% of cases), followed by the traditional *Cannonau wine*, which is consumed more than 1–2 times/week (nearly 80% of cases), *Minestrone*, which is consumed more than 1–2 times/week (nearly 76% of cases), *Culurgiones*, with at least 2–3 servings a month (91% of cases), and *myrtle liqueur*, consumed at least once a week (50% of cases).

#### 3.1.5. Correlation between Frequency of Food Consumption and Anthropometric Measurement

Correlations between the frequency of food consumption and anthropometric measurements are reported in the Appendix A. In the Nicoya subgroup none of Spearman’s coefficients were statistically significant; in the Sardinian subgroup, a significant correlation was found between meat consumption and body weight (ρ = 0.330) and meat consumption and circumferences of the arm (ρ = 0.466) and calf (ρ = 0.513). A positive correlation was observed between potato consumption and waist circumference (ρ = 0.373).

#### 3.1.6. Correlation between Food Consumption and Health Status

The correlation between eating habits and some of the health status indicators measured in the elderly from Nicoya and Ogliastra are listed in Table 4. Statistically significant positive correlations were found in both populations between milk consumption and functional ability (Nicoya: ρ = 0.268 for BADL and ρ = 0.466 for IADL; Sardinia: ρ = 0.214 for BADL and ρ = 0.471 for IADL). Interestingly, this association was observed in both sexes, although it was statistically significant only for IADL (Nicoya: ρ = 0.626 for male and ρ = 0.346 for female; Ogliastra: ρ = 0.388 for male and ρ = 0.205 for female). Functional capacity (BADL score) was positively correlated with sweets consumption (ρ = 0.285) in the Nicoya population. A positive correlation was found in Ogliastra between meat consumption and IADL score (ρ = 0.287). Conversely, a negative correlation was detected between legumes consumption and self−rated health score (ρ = −0.264), and IADL in both populations.

## 4. Discussion

In the present study, an investigation of dietary habits was conducted in two of the longest-lived populations belonging to the LBZs [17,23]. The analysis was done separately for the two sexes, in consideration of the diversity related to the different roles for men and women in these traditional societies, which are more prominent than in sophisticated post-modern societies. The social profile detected in the population of Nicoya was in agreement with a report by Madrigal-Leer et al. [21] and somewhat similar for the Ogliastra community [5]. Both are poor rural populations characterized by a remarkable history of long-standing isolation, not only in geographical terms but also genetic and cultural terms, which have been well documented in the literature (for Nicoya see [36,37], for Ogliastra, see [38]), although a discussion on the potential relationship between their genetic makeup and longevity is beyond the scope of this work. Although both areas are considered low-income regions compared to the rest of their respective countries, there is a difference of –56% in the average per capita income between Nicoya and Ogliastra. However, we must take into account that this is a mere indicator of “objective poverty” and does not include the subjective aspects (self-perceived poverty) [39]. Indicators of air pollution are lower in the two locations compared to the rest of their countries, but that of soil, in particular from pesticides, is not negligible due to the intense agricultural activity [40].

The analysis of anthropometric data shows that the elderly of Nicoya were on average taller and leaner than their Sardinian peers, in accordance with a previous observation [41]. Moreover, in this article, Nicoyans were reported to be taller compared with the rest of the country. This outstanding height was attributed by the authors to a healthier growth of this population in the early stages of body development. On the other hand, the shorter stature of the Sardinian elderly cannot be attributed to insufficient nutrition during childhood and adolescence, but rather to a higher frequency of short-statured genetic variants [42].

The analysis of the consumption frequencies of common foods in the two populations showed that cereals were the most frequently consumed food, with few differences between the two populations and the two sexes, followed by legumes and fruit, although their intake was significantly lower in the Ogliastra population. Interestingly, in the Nicoya subgroup there was a variable proportion, between 3% and 6%, who never or rarely ate this food. The salad consumption was around 1–2 servings a week in most interviewees and, although similar in the two populations, no Sardinian participants completely abstained from it, while among Nicoyans the percentage of those who did not eat salad was up to 15%. Furthermore, potato consumption was high in both populations: At least 1–2 servings per week in 50% of study participants with a 60% peak among Sardinian males. This seems in contrast with the alleged harmful health effect of these tubers, resulting from their high glycemic load potentially increasing the risk of diabetes [43,44]. However, these negative aspects are mostly related to the cooking method, especially frying [45]. Instead, in both populations, potatoes are mostly boiled and especially seasoned with fats that are able to reduce their glycemic index.

On the basis of these findings we can conclude that the diet of the two elderly populations is essentially “plant-based” [14], although a non-negligible consumption of animal-derived foods was also recorded. Half of the individuals were meat-eaters at a frequency not less than of 3–5 servings per week, and a quarter of Sardinian and Nicoyan males consumed meat almost daily. Contrary to what has been reported recently in Nicoya centenarians [22], the most consumed type of meat was pork and not chicken, although this was consumed, together with beef, in moderate quantities. Similarly, Sardinians consumed pork most frequently, followed by goat or lamb (data not shown). Of note, meat came from locally bred animals, and also ham, sausages, and salami were homemade. Therefore, this processed meat contains only salt and spices such as pepper and garlic without preservatives or chemical additives.

The non-negligible consumption of meat seems in contrast to what has been described in other populations [46]. Based on the results of the NHANES III study [47], the impact of meat consumption on health is age-dependent: Although meat products could be harmful before the age of 65, after this age the situation reverses, because meat may preserve the elderly from excessive loss of muscle mass, indirectly promoting longevity. Therefore, it is no wonder that it has been suggested to re-evaluate the role of meat in the daily diet, especially home-prepared meat from self-raised animals [48].

A similar consideration must be made for dairy food: About half of Nicoyan males and females consumed it every day, while the percentage exceeded 80% among Sardinians. They are always typical homemade products: In Nicoya fresh cheese or in the form of *cuajada*, in Sardinia mostly a fresh sour cheese (*casu ajedu*), although aged cheese was consumed as well. For a long time, dairy products have not been considered particularly healthy; however, a positive association between their consumption and longevity has been reported among Japanese centenarians [49], and it has been reported that they have a protective effect against dementia [50] and cardiovascular disease [51]. In this regard, it should be noted that, at least in Ogliastra, milk is almost never of bovine origin but comes from sheep and goat, which has less harmful effects on health and contains bioactive peptides that improve insulin activity [52]. More importantly, the high consumption of milk was significantly and positively associated with functional capacity of BADL and IADL scores. These findings were consistent between the two ethnic groups and the two sexes, making the association reliable.

Although fish consumption is associated with better health, and is strongly recommended by nutritionists, it was very scarce in the two long-lived populations. The frequency of fish consumption was confirmed to be very low among Sardinians, as previously reported [53], while it was a little higher among Nicoyans (more than 1–2 servings per week for only 10% of the total population). Another eating behavior shared by the two populations was the low consumption of sweets, since half of the Nicoyan males and two thirds of the Sardinian males ate less than 1–2 servings per month. Females from the two regions consumed slightly more.

Finally, coffee consumption was abundant among the study participants: More than 85% of the Nicoyans and more than 75% of the Sardinians drank coffee daily, a feature described also in Ikaria LBZ [54]. Long-term dose-dependent benefits of coffee have been reported including a reduced risk of type 2 diabetes, Parkinson’s and Alzheimer’s disease, alcoholic cirrhosis, and gout [55,56]. Moreover, several studies showed an inverse relationship between coffee consumption and mortality for any cause, albeit mild [57].

The interest in the relationship between human longevity and the diet dates back to the classical period and is reflected in the Hippocratic writings [58]. Only in modern times, however, the possibility of extending human lifespan by the adoption of a particular dietary regimen aroused the attention of scholars, as evidenced by the interest in famous fasters such as the Italian Alvise Cornaro in the Renaissance epoch [59]. At present, there is a growing body of research is focused on the relationships between eating habits and human longevity [46,47,48,49,50,51,52,53,54,55]. However, the media often spread news and statements not always based on evidence, making it urgent to accurately ascertain whether the information retrieved from long-lived populations is reliable and can provide guidance and recommendations generalizable to the broader population. Despite the difficulty in comparing the eating habits of populations geographically distant, and whose ethnic origins and evolutionary history are divergent, this effort has the advantage of strengthening the study’s findings, to a greater extent than an analysis performed on a single population, where the results could simply be due to chance.

Nonetheless, our study has some limitations. Firstly, the cross-sectional study design does not allow to determine cause-effect relationships, but only associations. Secondly, the FFQ used for the analysis, although widely used in many epidemiological studies, did not allow us to accurately estimate the calorie intake and is not able to ascertain any sort of calorie restriction. Moreover, being a “snapshot” of the current state of nutrition, our investigation lacks historical dimensions. However, the correlation between diet and health status in the LBZ research is a novelty and lays the foundations for future, more detailed studies on a greater number of variables collected. Finally, the large number of participants in our study ensured adequate statistical power to obtain reliable estimates.

## 5. Conclusions

In conclusion, the study of the oldest people of the Nicoya peninsula compared to those from Ogliastra, confirms that a plant-based diet, integrated with an adequate supply of animal proteins, represents the ideal way to maintain an optimal state of health and long-lasting functional and physical capacity and anthropometric indices, so as to ensure excellent quality of life during the aging process. Our data support the hypothesis that a relatively small intake of animal products, in addition to a basically plant-based diet, can be beneficial, for example avoiding or delaying sarcopenia and the subsequent onset of functional disability, as documented in the long-lived Sardinian population [51]. This assumption implies that a particular diet is not beneficial in absolute terms but may exert positive or negative effects in relation to the age of the subjects and the specific needs of the individual. However, the results of our study are preliminary and further analyses will provide more detailed information regarding the correlations examined.

## Figures and Tables

**Figure 1 nutrients-12-01621-f001:**
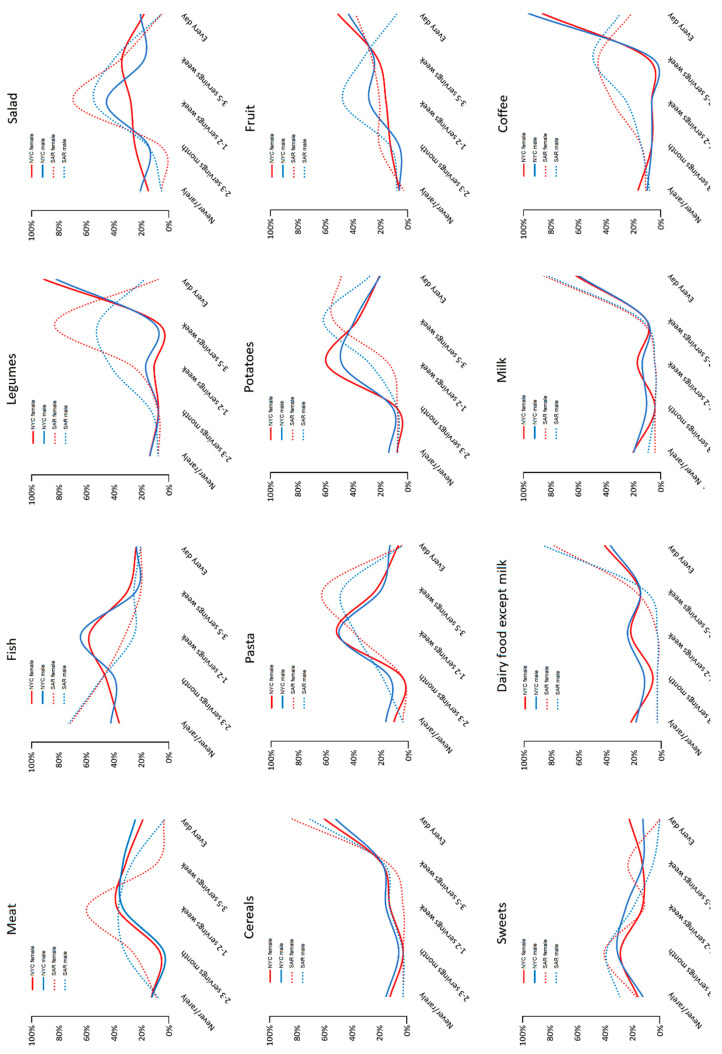
Frequency of consumption of common foods in participants from Nicoya (continuous lines) and Ogliastra (dotted lines), with male (blue lines) and female (red lines) highlighted.

**Table 1 nutrients-12-01621-t001:** Sociodemographic characteristics of the subjects from Nicoya and Ogliastra recruited for the study.

Variables	Nicoya		Ogliastra		
	Female (n = 29)	Male (n = 31)	Female (n = 89)	Male (n = 61)	*p*-Value
Age at recruitment Mean ± SD ^1^ Range	99.3 ± 5.7 87–109	96.5 ± 6.3 80–106	94.2 ± 3.3 90–101	93.4 ± 4.3 90–101	0.001 (F) 0.019 (M)
Marital status Single Married or free union Widowed Divorced	2 (6.9%) 7 (20.7%) 18 (62.1%) 2 (6.9%)	2 (6.5%) 15 (45.2%) 13 (41.9%) 1 (3.2%)	18 (20.2%) 21 (23.6%) 50 (56.2%) 0 (0.0%)	3 (4.9%) 10 (16.4%) 47 (77.0%) 1 (1.6%)	0.001 (F) 0.001 (M)
Education (years)	3.1 ± 2.6	3.5 ± 2.9	5.4 ± 2.1	3.2 ± 1.7	<0.001 (F) 0.596 (M)
Living conditions Owned or leased houses Nursing home	28 (96.6%) 1 (3.4%)	29 (93.5%) 2 (6.5%)	88 (98.8%) 1 (1.2%)	60 (98.4%) 1 (1.6%)	0.399 (F) 0.219 (M)
Smoke Never Former Current	24 (82.8%) 5 (17.2%) 0 (0.0%)	7 (22.6%) 21 (77.7%) 3 (9.7%)	87 (97.7%) 2 (2.2 %) 0 (0.0%)	8 (13.11%) 52 (85.2%) 1 (1.63%)	0.003 (F) 0.245 (M)

^1^ SD, standard deviation.

**Table 2 nutrients-12-01621-t002:** Anthropometric parameters in 210 Nicoyan and Sardinian participants.

Variables	Nicoya		Ogliastra		
	Female (n = 29)	Male (n = 31)	Female (n = 89)	Male (n = 61)	*p*-Value
Body weight (kg)	44.9 ± 9.3	58.3 ± 13.3	47.8 ± 8.5	59.6 ± 10.8	n.s.
Body height (cm)	149.0 ± 8.2	161.6 ± 6.9	143.5 ± 9.0	156.1 ± 13.2	0.002 (F) 0.010 (M) ¹
Body mass index (kg/m²) <18.0 18.0–24.9 25.0–29.9 ≥ 30.0	8 (27.6%) 19 (65.5%) 2 (6.9%) 0 (0.0%)	3 (9.7%) 21 (67.7%) 6 (19.4%) 1 (3.2%)	9 (10.0%) 49 (55.0%) 30 (33.7%) 1 (1.3%)	7 (11.4%) 27 (44.2%) 18 (29.5%) 9 (14.9%)	0.011 (F) 0.135 (M)
Waist circumference	86.9 ± 8.9	91.6 ± 13.2	101.9 ± 12.7	99.8 ± 12.3	0.001 (F) 0.005 (M)
Knee-floor distance (cm)	43.1 ± 2.4	46.3 ± 1.9	35.3 ± 4.2	38.8 ± 7.2	0.012 (F) 0.016 (M)
Average arm circumference (cm)	23.9 ± 3.5	24.7 ± 3.7	23.9 ± 4.6	26.1 ± 3.7	n.s.
Average calf circumference (cm)	28.3± 3.8	30.0 ± 4.4	29.6± 4.3	31.6 ± 5.1	n.s.

^1^ Female vs male (sex-specific comparison).

**Table 3 nutrients-12-01621-t003:** Basic activities of daily living and instrumental activities of daily living recorded among 210 Nicoyan and Sardinian participants.

Variables	Nicoya		Ogliastra		
	Female (*n* = 29)	Male (*n* = 31)	Female (n = 89)	Male (*n* = 61)	*p*-Value
Self-rated health	2.66 ± 1.29	3.06 ± 0.77	2.34 ± 0.95	4.16 ± 1.78	0.125 (F) 0.095 (M)
*BADL* Severe disability Moderate disability	8 (27.6%) 21 (72.4%)	4 (12.9%) 27 (87.1%)	22 (24.7%) 67 (75.3%)	9 (14.7%) 52 (85.2%)	0.758 (F) 0.810 (M)
*IADL* Severe disability Moderate disability	6 (20.7%) 23 (79.3%)	3 (9.7%) 28 (90.3%)	19 (21.3%) 70 (78.7%)	7 (11.5%) 54 (88.5%)	0.320 (F) 0.415 (M)

^1^ Women vs men (sex-specific comparison).

**Table 4 nutrients-12-01621-t004:** Correlation between frequency of consumption and health status in the elderly of Nicoya and Ogliastra.

Variables	Nicoya			Ogliastra		
	Self-Rated Health	BADL	IADL	Self-Rated Health	BADL	IADL
Meat	−0.085	0.080	0.003	−0.004	0.267	0.287 *
Fish	−0.242	0.091	−0.019	−0.147	0.253	0.162
Legumes	−0.264 *	−0.110	−0.253	−0.187	−0.238	−0.332 *
Salad	−0.032	0.084	−0.050	0.051	0.120	0.032
Cereal	−0.223	0.072	−0.070	−0.152	−0.036	−0.227
Pasta	−0.191	−0.042	−0.156	−0.067	−0.125	−0.163
Potato	−0.314	−0.166	−0.286	−0.234	−0.177	−0.113
Fruit	0.029	−0.067	−0.092	0.030	−0.268	−0.226
Sweet	0.016	0.285 *	0.195	−0.092	0.215	0.089
Dairy food except milk #	0.055	0.095	0.018	0.149	0.054	0.112
Milk	0.212	0.268 *	0.466 **	0.080	0.214 *	0.471 *
Coffee	−0.061	0.179	0.114	0.148	0.106	0.126

* *p* < 0.05; ** *p* < 0.01; ^#^ includes acid soft cheese.

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
