# Peer review of "Dietary Habits, Anthropometric Features and Daily Performance in Two Independent Long-Lived Populations from *Nicoya peninsula* (Costa Rica) and *Ogliastra* (Sardinia)"

_nutrients, 2020, doi:10.3390/nu12061621_

Round 1

Reviewer 1 Report

The research shows some interesting results about Blue Zones inhabitants' food habits, that could be of intested to understand the relationship between food, aging and well being. It is clearly written and methodology is appropiate to answer the planned objectives.

Anyway I have some not-important concerns and questions I would like to ask the authors 

1. As authors say, both study populations seem really similar in socioeconomic characteristics, but "poor rural environment" can be very different in Europe or in  Central America. Do you have any information about average earnings in both populations? it is not an important matter, but it will help to have a better image of target groups

 2. the difference in the heigth and leg-length catch mi attention, Nicoya's population being the most tall, is this a characteristic also seem in the youngs living in same area?? maybe this fact deserves longer discussion, as heigth is directly related with early undernutrtion.

3. lines: 299-302 :  "The salad consumption was around 1‒2 servings a week in most interviewees and, although similar in the two populations, none of Sardinian participants completely abstained from it, while among Nicoyans the percentage of those who ate salad was up to 15%." I don't understand this statement. Do you mean   "the percentage who did not eat salad was up to 15%??" 

4. Conclusions; lines 379-382 : "Our data supports the hypothesis that a progressively increased intake of animal products, as age advances, can be beneficial, for example delaying sarcopenia and the subsequent onset of functional disability, as documented in the long-lived Sardinian population. This statement cannot be concluded from this research as it is a cross-sectional research and you don't have any information about the past  intake of meat of the participants. Maybe it has been always high 

Author Response

Reviewer #1

The research shows some interesting results about Blue Zones inhabitants' food habits, that could be of intested to understand the relationship between food, aging and well being. It is clearly written and methodology is appropiate to answer the planned objectives.

Anyway I have some not-important concerns and questions I would like to ask the authors

  1. As authors say, both study populations seem really similar in socioeconomic characteristics, but "poor rural environment" can be very different in Europe or in Central America. Do you have any information about average earnings in both populations? it is not an important matter, but it will help to have a better image of target groups

Reply: we sincerely thank the reviewer for raising an extremely important point. We agree that poverty in Central America cannot be easily compared with that of Sardinia; moreover, Nicoya is still in a phase of economic transition whereby the average per capita income increases every year, while Sardinia is more stable and is even experiencing a slight economic decline. In a previous study we reported the local income for Nicoya and Ogliastra ($8,700 and $19.872, respectively), although the article was published in 2013, up to now a significant difference between average incomes was not registered. In the revised version of the manuscript we added the following sentence “Although both areas are considered low-income regions compared with the rest of their respective mother countries, there is a difference of  56% in the average pro capita income between Nicoya and Ogliastra. Howewer, we must take into account that this is a mere indicator of "objective poverty” and does not include the subjective aspects (self-perceived poverty) [39]. Pag 10, lines 304-307. The new reference no. 39 was also added.

  1. the difference in the heigth and leg-length catch mi attention, Nicoya's population being the most tall, is this a characteristic also seem in the youngs living in same area?? maybe this fact deserves longer discussion, as heigth is directly related with early undernutrtion.

Reply: This is also a very interesting question. There is a big controversy regarding the relationship between body stature and longevity. In general, being taller is associated with increased longevity (Ezzati et al., 2016) and Costa Ricans are taller than other populations of the Andean Latin America. Moreover, the elderly of Nicoya are taller than the rest of the country population, as reported by the CRELES study, and it has been suggested that they had healthier growth in the early stages of life (Rosero-Bixby, 2013). In fact, between Nicoyans and the other Costa Ricans the body height gap was the second most important difference, following depression. This phenomenon is opposite of that recorded in the Sardinia population, where in the Ogliastra area stature was shorter than in the rest of the island. However, even a century ago children and adolescents were not malnourished albeit they suffered from infectious diseases such as malaria, suggesting, beside nutrition, an influence of the genetic background on stature. Therefore, in the discussion of the revised manuscript we added the following sentence "The analysis of anthropometric data shows that the elderly of Nicoya were on average taller and leaner than the Sardinian peers, according with previous observation [41]. Moreover, in this article Nicoyans stature was reported to be taller compared with the rest of the country. This outstanding height was attributed by the authors to a healthier growth of this population in the early stages of body development. On the other hand, the shorter stature of Sardinian elderly, cannot be attributed to insufficient nutrition during childhood and adolescence, but rather to a higher frequency of short-statured genetic variants [42]". Page 10 lines 311‒317.

  1. lines: 299-302 : "The salad consumption was around 1‒2 servings a week in most interviewees and, although similar in the two populations, none of Sardinian participants completely abstained from it, while among Nicoyans the percentage of those who ate salad was up to 15%." I don't understand this statement. Do you mean "the percentage who did not eat salad was up to 15%??"

Reply: Sorry for the mistake. This was corrected in the revised version (Page 10, line 314).

  1. Conclusions; lines 379-382 : "Our data supports the hypothesis that a progressively increased intake of animal products, as age advances, can be beneficial, for example delaying sarcopenia and the subsequent onset of functional disability, as documented in the long-lived Sardinian population. This statement cannot be concluded from this research as it is a cross-sectional research and you don't have any information about the past intake of meat of the participants. Maybe it has been always high

Reply: Although that statement was made has a hypothesis, we agree with the reviewer that it cannot be the valid deduction from a cross-sectional study. In the revised version of the manuscript we changed it as follows "Our data supports the hypothesis that a relative intake of animal products, in addition to a basically plant-based diet, can be beneficial, for example avoiding or delaying sarcopenia and the subsequent onset of functional disability, as documented in the long-lived Sardinian population [50]. ". Page 12, line 392-395.

Reviewer 2 Report

The manuscript is interesting and merit more research.

I found it interesting as the authors are comparing two groups of population from different countries/ continents.  Have they considered the environmental and geographical differences? Is it realistic to compare, have the authors used a statistical method to eliminate these confounding factors?

Methods- Authors used FFQ and certain anthropometric data such as BMI, waist and limbs circumferences in addition to BADL and IADL.  What are the rationales for selecting these parameters only?

Results- It looks like both groups are havening similar amount of plant-based foods (about 60-80%). Right?  So, we cannot really say one was the highest?  

From dairy foods, how come only milk was selected. How about yogurt, cheese?

Has the total calorie intake considered in their diet analysis and calculation?

Conclusion- Diet (plant-based, lower fat?)  is one of the factors that contributes to longevity.

How about physical activity, environment (less pollution), and genetic?  How should we address these?

The manuscript can be improved if some clarification to the above questions will be given.

Author Response

Reviewer #2

The manuscript is interesting and merit more research.

I found it interesting as the authors are comparing two groups of population from different countries/ continents.  Have they considered the environmental and geographical differences? Is it realistic to compare, have the authors used a statistical method to eliminate these confounding factors?

Reply: We thank the reviewer for his/her insightful observation. The number of environmental and geographical factors between the two populations under study is so high that it would not have been possible to take them into account. Even if all these variables were collected, the number of these super elders is so low that any statistical analysis would then be underpowered! We are aware that geographical, environmental and, above all, historical factors may have heavily shaped the dietary structure of the two populations; however, an appropriate statistical approach would also have required a different study design. In this preliminary work exclusively focussed on basic aspects of nutrition we wanted to highlight, essentially dietary habits  analogies of two populations, belonging to the longest-lived populations in the world. In the revised manuscript we added the following statement “Indicators of air pollution are lower in the two locations compared to the rest of their countries, but that of soil, in particular from pesticides, is not negligible due to the intense agricultural activity [40] ".Page 10 lines 308‒310.

Methods- Authors used FFQ and certain anthropometric data such as BMI, waist and limbs circumferences in addition to BADL and IADL.  What are the rationales for selecting these parameters only?

Reply: the choice was dictated by a number of criteria. First of all, these tools are widely used in the literature for their accuracy and, more importantly, are objective, i.e. independent from any subjective assessment. In addition, in the specific setting, to use all available instruments to measures all parameters may be counterproductive. The family of the oldest old subjects accepted their parents to be interviewed at home for free. This required by the investigator a behaviour discreet and sensitive, avoiding to cause excessive stress to the interviewee and to cause excessive waste of time for family members. For this reason, we choose to not include the assessment of cognitive status and symptomatic depression among the variables, and to restrict the analysis to objectively measurable functional markers. In the revised manuscript we added the following sentence “The number of variables related to functional status was restricted to the functional ability in every daily life which has a better relevance for the quality of life". Page 4, line 149-150.

Results- It looks like both groups are havening similar amount of plant-based foods (about 60-80%). Right?  So, we cannot really say one was the highest? 

Reply: Overall this is corrected, however there were some differences among consumption frequencies of plant-based foods, as specified in the discussion section

From dairy foods, how come only milk was selected. How about yogurt, cheese?

Reply: we apologize, but in Figure 1 and in Table 4 with the terms "mature cheese" or "aged cheese" we wanted to indicate the categories of dairy food, that included soft cheese or sour soft cheese. The "milk" category was kept distinct from that of cheese. In the revised manuscript was only used the term “dairy food” and in the footnote we specified that it also included the fermented, acidic products of milk.

Has the total calorie intake considered in their diet analysis and calculation?

Reply: The Short FFQs do not measure total energy intake. This is an intrinsic limitation to the tool used.

Conclusion- Diet (plant-based, lower fat?)  is one of the factors that contributes to longevity.

Reply: This is a stimulating question but the cross-sectional design of our study did not allow to answer it. Besides, there is a big controversy on the relationship between plant-based diets and longevity and attempting even to get into this thorny problem would have triggered endless controversies. Consequently, our discussion is prudently limited to a few conjectural considerations.

How about physical activity, environment (less pollution), and genetic?  How should we address these?

Reply: The role of physical activity in the oldest old has been addressed in Sardinia (Pes et al., 2018), the role of pollution in Nicoya and Sardinia (de la Cruz et al., 2014) and genetics in Nicoya (Rehkopf,et al. 2018). For example, it has been reported that Nicoyans have significantly longer telomeres and significantly higher levels of DHEAS suggesting a lower level of stress. Given the amplitude of these topics, an in-depth discussion of them would have quickly led the manuscript off topic and to unduly length. We only added the following statements “The leukocyte telomere length, considered as an indicator of longevity, is higher among Nicoya’s inhabitants [19], however this trait also depends on factors related to lifestyle, such as stress and daily physical activity, and does not exclusively reflect the individual genetic makeup [20]", page 3, lines 99‒102; “In addition, genetic association studies have been performed on Sardinian oldest old, using markers notoriously associated with longevity. However, in terms of frequency, none of these markers have been shown to be significantly different from that of the general Sardinian population [28]”. Page 3, lines 122-124.

The manuscript can be improved if some clarification to the above questions will be given.